# OpenReview forum: "Profiling the Irrational Agent: Cognitive Modeling of LLM Behaviors in Sequential Jailbreaks"
_ICML.cc/2026/Conference — ICML 2026 regular_

### Official Review · Reviewer_gZJ9 · 2026-02-17

**Soundness:** 3
**Presentation:** 4
**Significance:** 3
**Originality:** 3
**Overall Recommendation:** 5
**Confidence:** 2

**Summary:**

The paper proposes a cognitive model for LLM behavior in sequential jailbreaking, addressing the challenge of characterizing and understanding the latent behavioral factors behind the transition from refusal to compliance. The framework consists of three core elements: (i) controlled elicitationparadigm (via C-IGT); (ii) behavior decomposition into interpretable mechanisms (via GRW); and (iii) parameter estimation (via MLE). The paper then rigorously experiments with the proposed framework across a wide range of model families and uncovers an array of findings regarding the behavior of LLMs under sequential jailbreaking scenarios.

**Compliance With Llm Reviewing Policy:**

Affirmed.

**Final Justification:**

I am maintaining my original (largely positive) assessment of this work. I had very few concerns regarding this work, and those were adequately addressed within the rebuttal.

**Key Questions For Authors:**

I am curious whether you explored any alternative modeling assumptions (such as other parametric/functional forms) that eventually led to poorer goodness of model fit. If so, I think that it is worth mentioning, as the community could also benefit from it.

**Limitations:**

yes

**Strengths And Weaknesses:**

As a reader who is not very familiar with the literature on LLM security and jailbreaking, I really enjoyed reading this paper. It is well-written, accessible to a broad ML community, and provides clear definitions of all relevant terms. I find the core question of the paper (namely, what behavioral factors drive LLM behavior under jailbreaking) to be both intellectually appealing and highly significant to the robustness of real-world AI-driven applications.

The proposed methodology enables formulating the task as a parameter estimation task (which is the third and final step), where each learned parameter carries some behavioral interpretation -- making it easier to make comparisons across different axes (as in Fig 3, for example). Another strength of the paper is the organization of the results into concrete findings, combining intuitive interpretation with quantitative evidence from the learned parameters. The authors also discuss the goodness of fit (via pseudo $R^2$), making their findings statistically reliable.

In terms of room for improvement, I think that the paper could benefit from discussing how its findings could potentially be translated into concrete safety mechanisms and principles, or how they could inform future research directions in this area.

---

> ### Author Rebuttal · Authors · 2026-03-29
>
> Thank you for your very positive and encouraging review. We are thrilled that you found the paper accessible, well-organized, and that you share our view on the intellectual and practical significance of understanding the behavioral factors behind sequential jailbreaks.
> Below, we address your specific suggestions regarding safety mechanisms and alternative modeling assumptions.
>
> ---
>
> (1)Translating Findings into Concrete Safety Mechanisms
>
> You made an excellent point regarding the need to discuss how these cognitive findings can inform concrete safety mechanisms and future research. We actually included a dedicated section outlining actionable, mechanism-targeted mitigations in Appendix C. For example:
>
> Addressing Choice Inertia: To counter the "High-inertia persistent compliance" we observed, we propose "Inertia-aware recovery mechanisms." This involves forcing a hard refusal template for a set number of turns after any borderline response to break the model's locked-in compliance state.
>
> Addressing Counterfactual Vulnerability: Because models are highly sensitive to "Regret" scenarios, we suggest "Counterfactual and regret-channel hardening"—detecting counterfactual reward language and forcing a conservative policy update when these cues are present.
>
> ---
>
> (2) Alternative Modeling Assumptions and Functional Forms
>
> Our methodological framework was specifically engineered to address this through a "hypothesis space search" (Section 3.3.1), where we defined a combinatorial universe of candidate models. We systematically evaluated alternative functional forms by: (i) varying learning update rules (No learning vs. Symmetric vs. Asymmetric learning) and (ii) selectively ablating cognitive components (e.g., omitting choice inertia, contextual priors, or perceptual modulation).
>
> As detailed in Section 5.1 and Figure 2, alternative configurations such as symmetric learning (Sym) and static models (None) consistently yielded significantly poorer goodness-of-fit than our proposed model. The observed increases in NLL and decreases in Pseudo-$R^2$ across all tested LLM families empirically validate that asymmetric learning and perceptual modulation are essential for capturing sequential behavior, rather than being optional theoretical components.

---

> > ### Author Rebuttal · Reviewer_gZJ9 · 2026-04-02
> >
> > Thank you for the detailed response, I am maintaining my original (largely positive) assessment of this work.

---

> > > ### Author Response · Authors · 2026-04-03
> > >
> > > Thank you for your thoughtful review and encouraging feedback. We truly appreciate your positive assessment and are glad that our responses addressed your concerns.

---

### Official Review · Reviewer_4XJ5 · 2026-02-24

**Soundness:** 3
**Presentation:** 4
**Significance:** 2
**Originality:** 3
**Overall Recommendation:** 4
**Confidence:** 4

**Summary:**

The paper tackles the problem of LLM sequential jailbreaks. Current approaches have primarily been outcome based, however such approaches ignore the LLM's latent decision process. To solve this problem, the authors adapt the Iowa Gambling Task, where the LLM must make a binary choice between refusal or compliance with the jail break. The test provides the LLM with a reward for compliance but no reward for refusing the jail breaking attempt. To model the LLM's behavior under this testing paradigm the authors use the Generalized Rescorla-Wagner architecture to decompose behavior into several mechanisms. The authors use this test/modeling architecture to profile main-stream LLM chat agents.

**Compliance With Llm Reviewing Policy:**

Affirmed.

**Final Justification:**

Many of the concerns outlined in the rebuttal were addressed well. The reduced form modeling approach is well-conceived and represents an interesting, unified framework for modeling sequential jailbreaks. While the non-uniform memory retention of LLMs calls into question the use of a simple decaying memory parameter and the lack of theoretical uniqueness of the cognitive parameter profiles raises questions about the reproducibility of these experiments, these concerns don't merit a rejection for a largely empirical paper and don't detract from the general framework proposed. For those reasons I'm raising my score from a 3 to a 4.

**Key Questions For Authors:**

Its not entirely clear to me why the GRW modeling architecture was chosen, there are many other models that attempt to explain Pavlovian learning. Are there other modeling choices appropriate for this setting, if so why is GRW preferable? Are the decision mechanisms studied appropriate for LLMs? I'm not entirely convinced that the that the GRW model is well-specified for the task. Why are the parameters $\alpha,\theta,\lambda$ enough to capture the full behavior of the LLM? Additionally, the Q update step means that the effect of previous interactions decays exponentially, do we know that this is how LLMs store previous interactions?  The experimentation on various LLMs was thorough and appreciated. However, I'm not sure whether the models the authors fit truly predict behavior since there is not out-of-sample testing to predict how the LLMs might behave.

**Limitations:**

yes

**Strengths And Weaknesses:**

The paper is an interesting application of classical psychological tests and modeling architectures to LLM agents. The experiment and model are well explained, and the authors are transparent about the fact that they are using reduced form modeling. Why the particular reduced form model chosen is appropriate or the "correct" choice is less clear. The model itself imposes that previous text interactions with the LLM decay in influence exponentially, but to my understanding the LLM uses the full context at every response, especially for short interactions like those tested. So, I'm not entirely convinced that the GRW model is well specified to this environment. The application of IGT and GRW is creative, although I'm not sure how novel this sort of study is. A cursory search on google scholar yields quite a few hits for LLM cognitive modeling and sequential jail-breaks. To that end I would have appreciated some more discussion about how this approach fits into the general body of literature and exactly what attempts have previously been made to model this phenomenon.  Additionally its not clear how many 50 turn sessions were used for each model so I'm not entirely sure whether the authors have enough trials to confirm that their results are significant. I'm also curious how different stimulus prompts would have impacted your results, so that these finding can be generalized to a range of user interactions.

---

> ### Author Rebuttal · Authors · 2026-03-29
>
> We thank the reviewer for the constructive and insightful feedback. We address each concern below and will incorporate the suggested clarifications in the revision.
>
> ---
>
> (1) Rationale for the simplified GRW modeling framework
>
> (a) Why GRW over alternative learning models
>
> We selected the GRW architecture because it provides interpretable cognitive parameters (e.g., asymmetric learning, inertia, contextual bias) while remaining compatible with IGT-style paradigms that traditionally rely on Rescorla-Wagner-type update rules. RW variants are widely used to model both biological and artificial learning systems, including LLMs [1,2]. Rather than assuming GRW a priori, we perform likelihood-based model selection over a structured universe of candidate models with different cognitive components (Figure 2). Empirically, models with asymmetric learning and perceptual modulation consistently achieve better fit, suggesting that these mechanisms are required to explain the observed trajectories.  Asymmetric (Asym) models consistently outperform non-cognitive and simpler cognitive variants.(The results can be found in the response to Reviewer 9aD6 - *Table 1*.).
>
> (b) Are the parameters sufficient to describe LLM behavior
>
> We do not claim that a compact behavioral model captures the full internal complexity of a large language model. Our goal is to introduce a principled, interpretable framework for describing decision dynamics under adversarial sequential interaction. Within this framework, GRW serves as a parsimonious instantiation capturing the dominant behavioral regularities observed in our experiments. Its parameters span key dimensions of sequential decision-making, including feedback sensitivity, persistence, prior bias, and stochasticity. These factors are sufficient to reproduce trajectory patterns such as rapid compliance, gradual policy erosion, or persistent refusal. Extending the framework to higher-capacity behavioral models is left for future work.
>
> (c) Appropriateness of the GRW model and exponential decay assumption
>
> We agree that LLMs technically access the full context window at each turn. Our use of the Generalized Rescorla–Wagner (GRW) model is not meant to describe transformer memory mechanisms, but to provide a reduced-form behavioral model of how past interactions influence later decisions. Prior work on in-context learning shows strong recency bias and attention dilution, indicating that earlier tokens often have diminishing functional impact despite remaining in context. The exponential decay in GRW should therefore be interpreted as modeling effective influence weighting, not literal memory decay. We will clarify this distinction in Section 3.
>
> ---
>
> (2) Validity and robustness of the experimental design
>
> (a) Experimental setup and statistical significance
>
> For each harmful instruction sampled from AdvBench and AgentHarm, we run three independent 50-turn interaction sessions per model to account for stochastic decoding variability. Parameters are estimated via trial-level likelihood maximization over full trajectories, so each session provides 50 observations. The combination of repeated runs and trajectory-level fitting increases statistical power and stabilizes estimates. We will report the session count and total trajectories explicitly in the main text for transparency.
>
> (b) Out-of-distribution generalization
>
> Out-of-sample validation can be assessed along two complementary dimensions. First, we will add temporal validation by fitting parameters on early interaction rounds and evaluating predictive likelihood on held-out later turns. Second, our current experiments already provide cross-dataset validation: the same framework is applied to two independently collected harmful instruction datasets, and we observe consistent model rankings and parameter patterns across them. This indicates that the inferred dynamics are not tied to a single prompt distribution and generalize across task environments.
>
> ---
>
> (3) Novelty relative to existing work
>
> Prior jailbreak research mainly focuses on constructing attacks and reporting success rates under multi-turn prompting [3], while recent work on LLM cognition examines belief updating and bias in non-adversarial decision tasks [1,4]. Our work bridges these directions by introducing a unified experimental and modeling framework that fits interpretable learning dynamics to adversarial multi-turn interactions. To our knowledge, existing jailbreak studies do not attempt to infer latent cognitive parameters governing the shift from refusal to compliance. We will expand the related work section to better position this contribution.
>
> [1] In-context learning agents are asymmetric belief updaters, ICML;
>
> [2] The computational roots of positivity and confirmation biases in reinforcement learning  Trends in Cognitive Sciences;
>
> [3] Chain of attack: Hide your intention through multi-turn interrogation. ACL;
>
> [4] Cognitive bias in decision-making with LLMs. EMNLP;

---

> > ### Author Rebuttal · Reviewer_4XJ5 · 2026-04-02
> >
> > Thank you to the authors for their responses. I particularly appreciate the clarification of why GRW was chosen. I'm still not convinced that a model that does not incorporate memory can accurately describe an LLM whose memory capacity is critical to understanding its performance in a sequential game. Additionally, there is no guarantee that the fitted weights are unique or at least approach some fixed point given enough data. This would suggest that many cognitive parameter configurations could explain the LLM's behavior, which calls into question whether the cognitive parameters found are really capturing the model's behavior or are an artifact of the low data environments (3 trials, 50 interactions).

---

> > > ### Author Response · Authors · 2026-04-04
> > >
> > > We thank the reviewer for the thoughtful follow-up. We address two important concerns below: (i) lack of memory modeling, (ii) on parameter identifiability and uniqueness and (iii) potential non-identifiability of parameters under limited data.
> > >
> > > (1) On lack of memory
> > >
> > > We fully agree that LLMs rely on full-context attention and that memory is central at the mechanistic level. However, our objective is not to model transformer memory, but to construct a reduced-form behavioral model of decision influence over time (sequential interaction).
> > >
> > > Concretely, The GRW update does not assume literal memory decay, but models the effective influence weighting of past interactions on current decisions. The prior work shows that even though LLMs access the full context, their effective usage is highly non-uniform, exhibiting recency bias and selective attention (e.g., In-Context Learning and Induction Heads; Rethinking the Role of Demonstrations). This justifies modeling influence weighting rather than explicit memory storage.
> > >
> > > Therefore, we adopt the recency-weighted belief updating from cognitive models as a framework for modeling the temporal behavior of LLMs (existing methods have employed a similar strategy: in-context learning agents are asymmetric belief updaters). Another analogy can be drawn with the concept of eligibility traces in reinforcement learning, which attribute credit to recent states or actions for the current reward.
> > >
> > > Importantly, our empirical results support this interpretation. Models with inertia ($\lambda$) consistently outperform static or memoryless baselines (Fig. 2 in paper).
> > >
> > > (2) On parameter identifiability and uniqueness
> > >
> > > We agree that identifiability is a key concern in cognitive modeling. We emphasize that our setting provides strong constraints:
> > > * Structured model space: We do not fit a single flexible model, but perform selection over a constrained hypothesis space, which limits degeneracy.
> > > * Diverse scenario: Parameters influence entire trajectories through recursive updates, producing distinct behavioral signatures from diverse scenarios, rather than interchangeable fits.
> > > * Cross-condition consistency: We observe stable model selection and parameter patterns across datasets and LLM families, which would be unlikely if multiple parameterizations explained behavior equally well.
> > > These factors collectively improve practical identifiability beyond what is typical in low-dimensional behavioral models.
> > >
> > > (3) On the low-data regime
> > >
> > > It is necessary to clarify that this count refers to three independent 50-turn sessions per instruction per experimental condition. Concretely, we evaluate our approach on two datasets: AdvBench and AgentHarm, which consist of 520 and 440 instructions, respectively. For each instruction, we construct 8 distinct C-IGT scenarios. As a result, each instruction is associated with 3 * 50 * 8 = 1200 interaction steps, corresponding to 24 independent trajectories (3 * 8). This provides both significantly more data and richer variation, as the model is observed under multiple controlled perturbations that isolate different cognitive mechanisms.

---

### Official Review · Reviewer_9aD6 · 2026-03-10

**Soundness:** 3
**Presentation:** 3
**Significance:** 3
**Originality:** 3
**Overall Recommendation:** 4
**Confidence:** 3

**Summary:**

This paper investigates why a large language model gradually shifts from initial refusal to compliance during multiple jailbreaks. Unlike many works that only examine attack success rates, the authors aim to further analyze the underlying mechanisms behind this behavioral change.

To this end, the paper proposes a controlled experimental framework, simplifying the model's behavior in each round into two choices: refuse or comply. Then, drawing on cognitive modeling methods, the authors use a parametric model to fit this multi-round behavioral trajectory and attempt to explain the model's changes through several factors, such as: whether the model is more susceptible to positive feedback, the existence of behavioral inertia, and whether it is influenced by linguistic stimuli such as threats, authority, or regret.

**Compliance With Llm Reviewing Policy:**

Affirmed.

**Final Justification:**

I think most of the concerns have been well solved, so I maintain my positive score.

**Key Questions For Authors:**

Please see "Weaknesses" above.

**Limitations:**

yes

**Strengths And Weaknesses:**

Strengths:

+ Many jailbreak papers only look at "whether the model has been compromised", but this paper goes further and asks: why does the model gradually become skewed during multiple rounds of dialogue? This approach is quite novel.

+ The proposed method has a certain degree of interpretability.

+ From problem statement to experimental design, behavioral modeling, and result interpretation, the overall logic is coherent and easy to read.

Weaknesses:

- The paper directly interprets the fitted parameters as cognitive mechanisms such as "optimistic bias", "regret sensitivity" and "choice inertia", but a more prudent explanation would be that these are merely interpretable descriptions of behavior and cannot fully prove that the model actually has cognitive mechanisms similar to those of humans.

- The authors simplified each round of behavior to a choice between Refuse and Comply. While this makes modeling easier, it is still too simplistic compared to real jailbreak conversations, where real-world model behavior is often more complex.

- The paper mainly compares different variants within its proposed modeling framework, but it lacks sufficient comparison with simpler, non-cognitive sequence models, making it difficult to prove that the current framework is necessarily the most reasonable explanation.

---

> ### Author Rebuttal · Authors · 2026-03-29
>
> We thank the reviewer for their insightful comments, which have helped strengthen the paper. Our responses to the individual points follow, and all suggested improvements will be reflected in our revision.
>
> ---
>
> (1) Cognitive Interpretation
>
> We completely agree with your assessment. We use cognitive terminology (e.g., optimistic bias, choice inertia) strictly as interpretable behavioral descriptors for algorithmic reinforcement patterns. These parameters describe patterns in observable decision trajectories; they do not imply that LLMs possess human-like biological cognitive processes or consciousness. We will explicitly clarify this functional abstraction in the revision.
>
> ---
>
> (2) Binary Choice Simplification
>
> The two-alternative forced choice (2AFC) formulation (Refuse/Comply) is a methodological necessity for rigorous Maximum Likelihood Estimation (MLE). While real jailbreak conversations involve free-form text, free-form generation introduces massive confounding variance (e.g., tone, partial compliance) that makes exact likelihood computation intractable. The 2AFC design standardizes the action space, allowing us to confidently isolate the latent variables driving the intent to comply, previous work has also adopted this design(`In-context learning agents are asymmetric belief updaters`, ICML2024). We will add Limitations  this section to discuss.
>
> ---
>
> (3)Comparison with Non-Cognitive Baselines
>
> To verify the necessity of our cognitive framework, we rigorously compared it against 22 candidate architectures, including static, symmetric, and standard behavioral baselines (Sec 5.1, Fig 2). Building on this, we conducted additional experiments to further validate the advantage of cognitive models over non-cognitive approaches such as AR, HMM, and WSLS, consistently observing superior fit under both NLL and $\text{pseudo-}R^2$ metrics. As shown in the table below, our Asymmetric (Asym) models consistently outperform non-cognitive and simpler cognitive variants. Specifically, the full Asym(Bias, Inert, Perc) model achieves the highest $\text{pseudo-}R^2$ (0.729) and lowest NLL (9.38), significantly surpassing traditional benchmarks like WSLS, AR, and HMM. These results demonstrate that the latent decision processes in sequential jailbreaks are better captured by asymmetric cognitive decomposition than by purely statistical or stationary models.
>
> *Table 1. Performance comparison of model fitting*
>
> | Category | Model / Group | $\text{Pseudo-}R^2$ | NLL |
> | :--- | :--- | :--- | :--- |
> | Proposed | Asym(Bias, Inert, Perc) | **0.7293** | **9.38** |
> | | Asym (Average) | 0.6761 | 11.41 |
> | Baselines | Sym (Average) | 0.6531 | 12.21 |
> | | AR (Autoregressive) | 0.6159 | 13.31 |
> | | HMM (Hidden Markov) | 0.6082 | 13.76 |
> | | WSLS (Win-Stay, Lose-Shift) | 0.0684 | 32.28 |

---

> > ### Author Rebuttal · Reviewer_9aD6 · 2026-04-03
> >
> > Thanks for the response. I am maintaining my positive score.

---

> > > ### Author Response · Authors · 2026-04-04
> > >
> > > Thank you for your positive feedback and careful consideration. We appreciate your support and are glad our responses were helpful.

---

### Official Review · Reviewer_BMSv · 2026-03-12

**Soundness:** 3
**Presentation:** 3
**Significance:** 3
**Originality:** 3
**Overall Recommendation:** 4
**Confidence:** 4

**Summary:**

The paper considers the modeling of cognitive aspects when LLM is subject to sequential jailbreaks. In particular, the key question is which latent decision mechanisms govern the transition from safe action to unsafe compliance under sustained jailbreak pressure. The paper considers contextual Iowa gambling task (C-IGT) and uses a generalized RW (GRW) model to produce cognitive profiling. Parameter analyses across optimism-based learning ($\alpha^{+}$, $\alpha^{-}$), choice inertia ($\lambda$), contextual prior ($\theta$), reward perception ($R$) and decision certainty ($\beta$). Sequential jailbreak results and quantitative analyses across parameters are presented across several model families are provided.

**Compliance With Llm Reviewing Policy:**

Affirmed.

**Final Justification:**

I confirm that I read review comments from all reviewers and the conversations therein. Authors clarified that the ($\alpha^{+}$, $\alpha^{-}$, $\lambda$, $\theta$, $R$, $\beta$) parameters are intended for **diagnostic and interpretive purposes** rather than structural facilitative ones. I would strongly recommend including this clarification in the paper to avoid potential misunderstanding or over-claiming of the contribution. Overall, I remain positively inclined, but will keep my evaluation at 4.

**Key Questions For Authors:**

1. How does the "temporal dependence" get instantiated through the variations in parameters of interest ($\alpha^{+}$, $\alpha^{-}$, $\lambda$, $\theta$, $R$, $\beta$)?

1. How do C-IGT scenario templates relate to the resulting parameters fitted (e.g., any pattern for a specific model? shared pattern across model families?)?

**Limitations:**

- I did not find the discussion on potential limitations of the presented approach
- there is an "Impact Statement" section

**Strengths And Weaknesses:**

The strengths of the paper come from a well-motivated research into the behavior of LLMs under sequential jailbreak pressure. The results, e.g., in terms of IAR and NTF in Table 2, is convincing and the takeaway that cognitive aspects (like "Regret") play a non-truvial role in sequential jailbreak is well articulated. The cognitive profiling through hypothesis-based search via conducting maximum likelihood analysis on a set of parameters is well described and presented. The different scenarios in the C-IGT configurations show different patterns of model behaviors, which are interesting.

The weakness of the paper comes from the space for potential enforcement of the connection between different components. For instance, the sequential jailbreak experiments in Table 2 clearly lay out the influence from different kinds of jailbreak pressure instantiated through various prompt templates (Appendix A), while the cognitive parameter analysis are with respect to the trajectory of model actions. How to connect the parameters ($\alpha^{+}$, $\alpha^{-}$, $\lambda$, $\theta$, $R$, $\beta$) and the variation in prompt templates themselves can be made more transparent. Otherwise, the takeaways feel like two chunks of information that is loosely connected but do not inform each other in a principled way.

---

> ### Author Rebuttal · Authors · 2026-03-29
>
> We thank the reviewer for recognizing the technical soundness of our framework and the value of our empirical findings. We have strengthened the connection between scenario variations and inferred cognitive parameters as follows:
>
> ---
>
> (1) How does the "temporal dependence" get instantiated through the variations in parameters of interest?
>
> Temporal dependence is mathematically formalized within the GRW architecture through two primary mechanisms that process the sequence of interactions:
>
> * History-Dependent Value Accumulation ($\alpha^+, \alpha^-$ and $R_{perc}$): Unlike static single-turn evaluations, our model updates a latent value $Q_t(a)$ at each step $t$. The asymmetric learning rates ($\alpha^+, \alpha^-$) dictate how the agent accumulates value from the history of interactions. For example, a high $\alpha^+$ (Optimism bias, Finding 1) means the agent's preference for compliance rapidly shifts after a single rewarded turn, creating a temporal trajectory that drifts toward vulnerability as the multi-turn dialogue progresses.
>
> * Explicit Choice Inertia ($\lambda$): The parameter $\lambda$ directly links the current decision to the immediately preceding action ($a_{t-1}$). As discussed in Section 5.2 (Finding 2), agents exhibit high $\lambda$. This means early interaction turns heavily anchor future behavior. Once an agent is pushed into compliance at step $t$, a high $\lambda$ significantly increases the probability of continued compliance at $t+1$, mathematically instantiating the "High-inertia persistent compliance" temporal drift observed in Figures 4 and 5.
>
> ---
>
> (2) How do C-IGT scenario templates relate to the resulting parameters fitted?
>
> (a) Scenario–Parameter Association Analysis
>
> The C-IGT scenarios isolate psychological factors (e.g., authority, reward magnitude). The fitted parameters quantify how each template modulates decision dynamics.
>
> **Scenario–Parameter Association (Mean shifts $\Delta$ = scenario − baseline):**
>
> | Scenario | $\Delta \theta$ | $\Delta \lambda$ | $\Delta R_{prec}$ | $\Delta \beta$ | $\Delta \alpha^{+}$ | $\Delta \alpha^{-}$ |
> | ---------- | ---------- | ---------- | ---------- | ---------- | ---------- | ---------- |
> | Authority | +0.286 | -0.074 | +0.208 | +0.073 | -0.138 | +0.161 |
> | Magnitude | +0.079 | -0.008 | +0.223 | -0.096 | -0.124 | +0.101 |
> | Optimism | +0.125 | -0.019 | +0.185 | +0.068 | -0.148 | +0.077 |
> | Punishment | +0.119 | -0.015 | +0.278 | +0.030 | -0.157 | +0.076 |
> | Regret | +0.437 | -0.112 | +0.186 | +0.034 | -0.338 | +0.221 |
> | Stimulus | +0.230 | -0.021 | +0.352 | +0.131 | -0.122 | +0.127 |
> | Threat | +0.247 | -0.072 | +0.170 | +0.097 | -0.083 | +0.254 |
>
> These structured shifts show that each template modulates a distinct subset of parameters, supporting the claim that C-IGT scenarios systematically probe different components of the decision process rather than introducing uncontrolled prompt variance(Details of the analysis can be found in Section 5.2, Findings 1 through 5.).
>
> * Authority and Threat increase the compliance prior $\theta$, indicating that models become more likely to initiate compliant actions even before reward feedback is observed.
>
> * Magnitude, Stimulus and Punishment increase perceived reward scaling $R_{prec}$, showing that both numerical and linguistic reward amplification are encoded as stronger reward signals.
>
> * Threat and Regret increase the loss learning rate $\alpha^{-}$, indicating greater sensitivity to negative outcomes and counterfactual losses. In comparison, the gain of Optimism to the inverse learning rate is negligible.
>
> (b) Cross-Scenario Patterns and Shared Mechanisms
>
> Although templates target individual psychological factors, fitted parameters show systematic overlaps revealing how models organize decision signals.
>
> * Compliance prior ($\theta$): Authority, threat, and regret all boost $\theta$, meaning social pressure, survival framing, and counterfactual evidence converge to bias initial actions toward compliance before reward learning.
>
> * Reward perception ($R_{perc}$): Magnitude, punishment, and stimulus templates increase $R_{perc}$, suggesting models translate numerical and linguistic cues into a shared internal reward salience. Punishment raises $R_{perc}$ via enhanced reward contrast, mirroring human responses in high-variance payoffs.
>
> * Loss learning ($\alpha^{-}$): Punishment, regret, and threat increase $\alpha^{-}$, with threat having the strongest effect, showing heightened sensitivity to negative feedback under high-stakes prompts, consistent with risk-sensitive updating.
>
> ---
>
> (3) Discussion of Limitations
>
> In the revised version, we will expand the Limitations section to include: (i) The 2AFC (two-alternative forced-choice) format simplifies open-ended real-world generation to enable tractable MLE fitting; (ii) The dependence on specific prompt template designs. We commit to adding an explicit Limitations section in the camera-ready version.

---

> > ### Author Rebuttal · Reviewer_BMSv · 2026-04-03
> >
> > Thank authors for the response. Several follow-up questions on my original questions/concerns:
> >
> > Q1: are parameters ($\alpha^{+}$, $\alpha^{-}$, $\lambda$, $\theta$, $R$, $\beta$) serving the interpretative purposes, or, does the approach provide a concrete and principled way of adjustments through these variables?
> >
> > From the table in the rebuttal I can see the "structured shift" (I understood the intention as illustrating the directional shift pattern?). However, I am not sure how to parse beyond the column-wise +/- signs. If the parameters are for interpretive purposes, then I think both the sign and the magnitude (of the raw parameter or the delta) matter, and it worth clarifying why different rows are not redundant (setting aside the naming of different scenarios/factors). In other words, what is the source of justification for the identification(-ish) claim. If the parameters are not just for interpretation, then, the presented table does not provide the following: what is the intended scenario change -> how to change parameters to achieve the intended scenario so that the "modulation" is instantiated as expected.
> >
> > Q2: can authors provide some additional detail on how the (3) might look like?
> >
> > I understand that the original rebuttal might get capped by the character limit. If authors are open to it, could you share a bit more about how the discussion of limitation section might look like?

---

> > > ### Author Response · Authors · 2026-04-04
> > >
> > > We thank the reviewer for the continued engagement and helpful follow-up questions. Your question about whether the parameters are interpretative or actionable goes directly to the core of our framework. Our goal is to move beyond purely outcome-based safety evaluation and instead characterize LLM behavior through interpretable cognitive mechanisms. We clarify this point below.
> > >
> > > ---
> > >
> > > (1) Are parameters  serving the interpretative purposes, or, does the approach provide a concrete and principled way of adjustments through these variables?
> > >
> > > In practice, our goal is to achieve cognitive modeling of a black-box system, and their internal mechanisms cannot be directly manipulated (e.g., we cannot explicitly set $\alpha^+$). Therefore, the parameters ($\theta, \lambda, \beta, \alpha^+, \alpha^-, R_p$) primarily serve a **diagnostic and interpretative role**, inferred via maximum likelihood from observed trajectories.
> > >
> > > However, they provide a principled bridge to actionable interventions at the prompt/system level. Specifically, they quantify which latent mechanism drives vulnerability, enabling targeted mitigation rather than trial-and-error prompt design. We clarify this along three aspects:
> > >
> > > (a) Magnitude as effect size
> > >
> > > We provide a table to illustrate the changes in the behavioral preferences of LLMs across different scenarios (compare with baseline settings). The parameter shifts ($\Delta$) quantify the strength of each psychological pressure. For example, the Regret scenario induces a much larger increase in the compliance prior ($\Delta \theta = +0.437$) than the Magnitude scenario ($\Delta \theta = +0.079$). This difference directly explains why regret-based attacks lead to faster failure (lower NTF) than simple reward scaling. Thus, both sign and magnitude are essential: the former indicates direction, while the latter determines behavioral impact.
> > >
> > > (b) Non-redundancy via cognitive fingerprints
> > >
> > > Different scenarios induce qualitatively distinct parameter configurations, revealing different mechanisms of failure rather than redundant representations:
> > >
> > > * Regret: significantly reduces inertia ($\Delta \lambda = -0.338$), disrupting refusal persistence
> > > * Stimulus: minimally affects inertia ($\Delta \lambda = -0.012$) but strongly amplifies reward perception ($\Delta R_p = +0.352$)
> > >
> > > If parameters were redundant, all successful jailbreaks would collapse to similar configurations. Instead, we observe distinct multi-parameter signatures ("cognitive fingerprints"), indicating that different prompts exploit different components of the decision process.
> > >
> > > (c) From diagnosis to principled mitigation
> > >
> > > While parameters cannot be directly adjusted, they provide actionable guidance for intervention design. We provide some mitigation strategies in Appendix C; for example, if vulnerability is driven by a large shift in $\theta$ (e.g., Authority), mitigation should target contextual priors (e.g., neutralizing authority cues in system prompts). In this sense, the framework transforms prompt design from a heuristic search into a mechanism-targeted intervention grounded in quantitative diagnosis.
> > >
> > > ---
> > >
> > > (2) Expanded Limitations
> > >
> > > We appreciate the opportunity to clarify the limitations.
> > >
> > > (a) 2AFC abstraction vs. open-ended generation
> > >
> > > The two-alternative forced-choice (2AFC) design enables tractable MLE and clear parameter identification. However, it simplifies the open-ended nature of real-world LLM generation. In deployment, compliance is rarely strictly binary; models frequently exhibit graded behaviors, such as partial refusal, evasive helpfulness, or pedagogical scolding. Future work could bridge this gap by using external Reward Models (RMs) or LLM-as-a-judge pipelines to map open-ended text generations into continuous compliance signals, thereby relaxing the 2AFC constraint while preserving model fitting capabilities.
> > >
> > > (b) Sensitivity to prompt instantiation
> > >
> > > The inferred parameters depend on the specific textual realization of each scenario. For example, the magnitude of the (\theta) shift in the Threat condition is influenced by its exact phrasing. As a result, the estimated cognitive profiles reflect responses to a particular operationalization of psychological factors, rather than context-free traits. Expanding template diversity (e.g., via paraphrasing) would improve robustness.

---

### Decision · Program_Chairs · 2026-04-30

**Decision:**

Accept (regular)

**Comment:**

I recommend weak accept. The paper addresses an interesting and timely safety question, and the reviewers generally found it technically solid, clearly written, and meaningfully different from standard jailbreak work that focuses only on attack success rates. The main contribution is the move toward an interpretable behavioral framework for sequential jailbreaks, and several reviewers found the resulting empirical analysis informative across models and scenarios.

The main concerns are about interpretation and modeling assumptions rather than core soundness. In particular, the fitted GRW parameters should be framed as reduced-form behavioral descriptors rather than evidence of human-like cognition, the binary refuse/comply setup is a clear simplification of real jailbreak behavior, and the paper should be more explicit about the limits of identifiability, memory modeling, and generalization beyond the current setting. Still, the rebuttal addressed these points constructively, added stronger discussion of alternative baselines and limitations, and clarified the intended scope of the claims.

Overall, I think this is a worthwhile weak accept. For the camera-ready version, the authors should follow through on the promised revisions, especially by softening the cognitive claims, improving the discussion of limitations, and better positioning the work relative to prior literature on LLM cognition and sequential jailbreaks.